

# Comparison of cryptobenthic reef fish communities among microhabitats in the Red Sea

Emily M. Troyer, Darren J. Coker and Michael L. Berumen

Red Sea Research Center, Division of Biological and Environmental Sciences and Engineering, King Abdullah University of Science and Technology, Thuwal, Saudi Arabia

## ABSTRACT

Knowledge of community structure within an ecosystem is essential when trying to understand the function and importance of the system and when making related management decisions. Within the larger ecosystem, microhabitats play an important role by providing inhabitants with a subset of available resources. On coral reefs, cryptobenthic fishes encompass many groups and make up an important proportion of the biodiversity. However, these fishes are relatively small, exhibit extreme visual or behavioral camouflage, and, therefore, are often overlooked. We examined the differences in fish community structure between three common reef microhabitats (live hard coral, dead coral rubble, and sand) using ichthyocide stations in the central Red Sea. Using a combination of morphological and genetic (cytochrome oxidase I (COI) barcoding) techniques, we identified 326 individuals representing 73 species spread across 17 families, from fifteen 1 m$^2$ quadrats. Fish assemblages in the three microhabitats were significantly different from each other. Rubble microhabitats yielded the highest levels of fish abundance, richness, and diversity, followed by hard coral, and then sand. The results show that benthic composition, even at a small scale, influences cryptobenthic communities. This study also provides new COI sequence data to public databases, in order to further the research of cryptobenthic fishes in the Red Sea region.

## INTRODUCTION

Habitat influences species abundances and distribution patterns in a range of ecosystems (*Venier & Fahrig, 1996*; *Warren et al., 2001*). Microhabitats on coral reefs, such as areas dominated by living hard corals, soft corals, rubble patches, macroalgae, or sandy areas, can offer a range of resources such as food and shelter for small fishes (*Beukers & Jones, 1997*; *Depczynski & Bellwood, 2004*; *Brooker, Munday & Ainsworth, 2010*; *Coker, Wilson & Pratchett, 2014*). Corals provide refuge spaces within the branches (*Robertson & Sheldon, 1979*), while the colony itself can provide shelter underneath for larger fishes (*Kerry & Bellwood, 2012*). It is expected that small and benthic-associated fishes would be influenced greatly by available habitats, and, therefore, understanding microhabitat requirements is essential for reef fishes.

Corresponding author
Emily M. Troyer,
emily.troyer@kaust.edu.sa

On coral reefs, there are many groups of fishes that are relatively small (< 50 mm), have a close association with the substrate, and have a cryptic nature (*Depczynski & Bellwood, 2003*). These fishes, termed "cryptobenthic reef fishes," can be behaviorally cryptic by seeking out cracks and crevices in the reef in which to hide, or they can be visually cryptic, having coloration that matches the substrate where they live (*Depczynski & Bellwood, 2003*; *Goatley & Brandl, 2017*). Fast-growing and with naturally short lifespans, these fishes are an important functional group on coral reefs (*Depczynski & Bellwood, 2006*). Microhabitat preferences have been well studied in blennies (*Wilson, 2001*; *Gonçalves & Faria, 2009*) as well as gobies (*Munday, Jones & Caley, 1997*; *Munday et al., 2002*). Their small size, coupled with small home ranges (*Luckhurst & Luckhurst, 1978*), has enabled them to make use of a multitude of different microhabitats.

Despite their functional importance, cryptobenthic fishes are difficult to sample due to their cryptic nature and, therefore, relatively understudied worldwide. When compared to other methods, visual surveys underestimate the number of cryptobenthic species on a reef (*Ackerman & Bellwood, 2000*; *Robertson & Smith-Vaniz, 2008*). Most surveys ignore them due to logistical and taxonomic difficulties. Thus, more targeted surveying techniques have recently been employed in order to gain a more accurate count of cryptobenthic reef fishes. Chemical ichthyocides, such as rotenone and clove oil, are an effective tool to collect small fishes, and reveal more cryptic species when compared to traditional visual surveys (*Brock, 1982*).

A majority of cryptobenthic fish studies have arisen from well-studied areas, such as the Great Barrier Reef and the Caribbean (see *Ackerman & Bellwood, 2000*, *2002*; *Harborne et al., 2012*; *Goatley, González-Cabello & Bellwood, 2016*), but an important region that is greatly understudied is the Red Sea (*Berumen et al., 2013*). The Red Sea is a unique environment, with higher average temperatures and salinity compared to other regions (*Edwards, 1987*). It is recognized as a biodiversity hotspot, with an estimated 14% of fish species in the Red Sea endemic to the region (*DiBattista et al., 2016*). Recent studies in this region show that cryptobenthic fish communities differ latitudinally, with distance from shore, and that habitat may be a driving factor (*Coker et al., 2018*).

This study aims to explore small-scale differences in the community composition of cryptobenthic reef fishes associated with three common reef microhabitats. Live hard corals, rubble patches, and sandy areas all represent habitats that cryptobenthic fishes are known to utilize (*Depczynski & Bellwood, 2004*; *Ahmadia, Pezold & Smith, 2012*; *Tornabene et al., 2013*).We hypothesize that cryptobenthic fish assemblages will significantly differ between these three microhabitats. The information gathered will begin to build a preliminary framework of Red Sea cryptobenthic fish ecology and provide much needed data, as well as vouchered samples, for these understudied fishes.

## MATERIALS AND METHODS

### Study site

Fishes were sampled in May 2017 during daylight hours from the southern portion of Al Fahal reef situated in the middle of the continental shelf (22°13.6558N, 38°58.1853E) (Fig. 1A). Sampling was conducted between 10 and 15 m depth (majority of stations

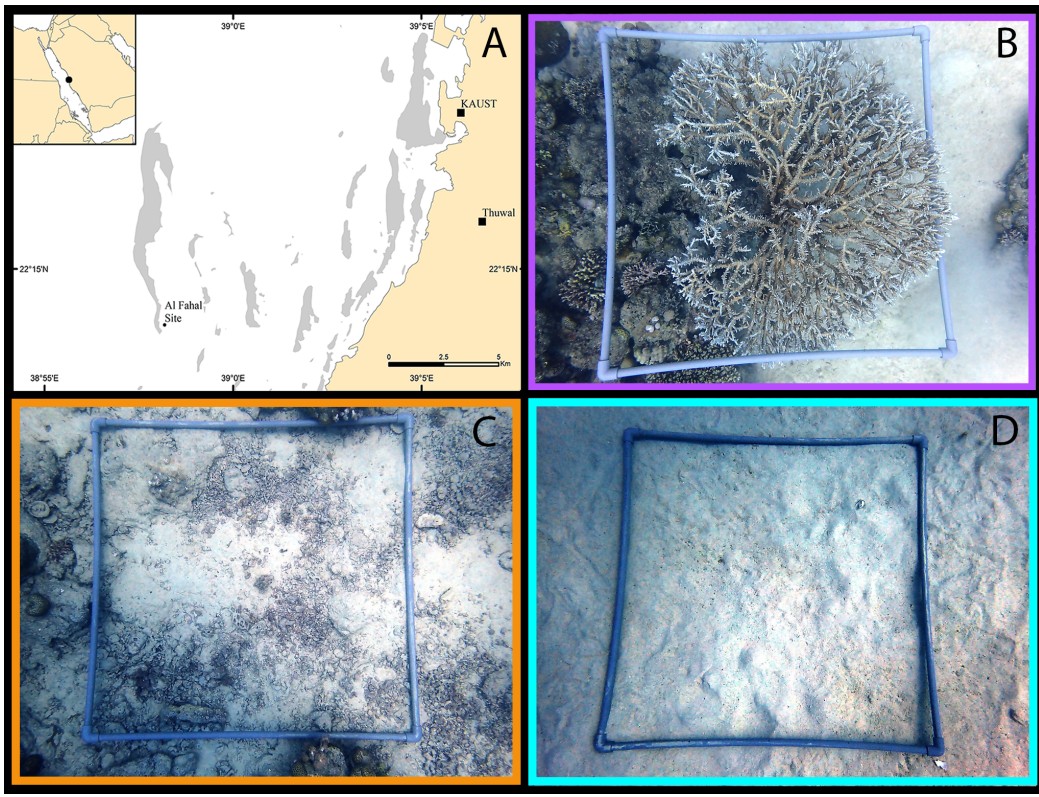

**Figure 1 Example microhabitat quadrats.** Examples of 1 m² quadrats sampled in the central Saudi Arabian Red Sea to assess cryptobenthic fish assemblages in each of the three microhabitat types: live coral (indicated by purple), dead coral rubble (orange), and sand (teal). Quadrats were sampled using rotenone at 10–15 m depth. (A) Map of study site, with reef habitat indicated in grey; (B) Coral quadrat; (C) Rubble quadrat; (D) Sand quadrat. (Photographs taken by EMT. Map created in ArcMap, version 10.3, by Michael Campbell using various mapping sources freely available through ESRI (Environmental Systems Research Institute, Redlands, CA, USA)).

between 12 and 13 m) (Table S1). This single reef and narrow depth range were selected to minimize any potential environmental variables, such as temperature, wave energy, current, and turbidity that could be introduced if sampling in varying parts of the reef (e.g., sides exposed to or sheltered from dominant wave action).

## Microhabitats

Quadrats (1 m²) constructed from PVC pipe (see Fig. 1) were placed onto a flat or gently sloping surface (<10°) of the reef in one of the three targeted microhabitats. Three microhabitat types were sampled: live hard coral, dead coral rubble, and sand (Figs. 1B–1D). For each microhabitat type, five replicate quadrats were sampled. Hard coral microhabitats were defined as having at least 70% of the survey quadrat covered with hard coral, with the remaining area composed of other microhabitat components, as seen from an aerial view. Tabular *Acropora* species were targeted for consistency. Rubble microhabitats contained at least 50% coverage of dead hard coral rubble, rocks, and/or empty shells. Sand microhabitats contained at least 90% coverage of sand with little or no rubble mixed in Table S1. Each quadrat was placed at least 1–2 m away from a neighboring

microhabitat type. Percent cover of each microhabitat category was calculated using a grid point system. A 10 ×10 grid (i.e., 100 total intercepting points) was overlaid onto a digital image of each quadrat. Each point in the image was categorized into the above microhabitat type to produce a total percent cover. Rugosity of the sample quadrat was measured as a linear distance using a 1-m chain (sensu *Risk, 1972*).

## Fish communities

An ichthyocide (rotenone) was used to collect fish due to its high success rate in targeting cryptic species (*Robertson & Smith-Vaniz, 2008*). A rotenone mixture was prepared by mixing 500 g of 4% rotenone powder (Consolidated Chemical Company, Dandenong South, Victoria, Australia) with 100 ml 96% ethanol, 250 ml liquid dishwashing detergent, and 100 ml water (adapted from *Ackerman & Bellwood, 2000*).

A 4 mm mesh net weighted down with chain was placed around the quadrat, enclosing the whole area. The rotenone mixture was gently squirted into the netted area until the whole area was covered. After waiting about 5 min for the rotenone to take effect, the net was removed and fishes were collected using hand nets and tweezers (following *Ackerman & Bellwood, 2000*). Because some fish were observed escaping the enclosed quadrat (either through the reef matrix or net) while the rotenone took effect, all deceased fishes directly around the sample area were collected for consistency. Water flow across the quadrats was minimal during ichthyocide deployment. Therefore, the effects were restricted primarily to within the sample area. However, we cannot discount some individuals outside the area succumbing to drifting rotenone. In these cases, the small number would have minimal influence in overall community composition. After collecting the visible fishes, three divers intensively searched within each quadrat by lifting up any loose rubble or debris where hidden fishes could have settled. Quadrats were searched until no new fishes were found for a period of 5 min. Larger predatory fishes that came too close to the quadrat were chased away before they could consume any of the asphyxiated fishes.

Immediately after the dive, collected fishes were placed into an ice slurry to preserve coloration for photography (within ~2 h). Once photographed, fishes were placed into individual labeled vials containing a solution of 96% ethanol for preservation. A tissue sample (pectoral or caudal fin) was collected from each individual for genetic analysis.

Fish sampling was done in accordance with the guidelines and procedures approved under the auspices of the King Abdullah University of Science and Technology (KAUST) Institutional Animal Care and Use Committee (IACUC) under approval number 17-04-004.

## Genetic fish identification

Tissue samples were cleaned with 96% ethanol and gently patted dry before placed into 96-well plates containing 100 μl 50 mM NaOH. DNA was extracted from the samples using the HotSHOT protocol (95 °C for 20 min, 4 °C for 10 min) (*Meeker et al., 2007*). After extraction, 10 μl of 1 M Tris–Hcl (10%) was added to each well and mixed with pipettes.

A total of 80 µl from each sample was transferred to a new plate for DNA amplification. A polymerase chain reaction (PCR) Qiagen Multiplex Mastermix containing Taq polymerase, dNTPs, MgCl$_2$, and reaction buffers was added to each well. Primers COI Universal Fish R2 5′ ACT TCA GGG TGA CCG AAG AAT CAG AA 3′ and F2 5′ TCG ACT AAT CAT AAA GAT ATC GGC AC 3′ (*Ward et al., 2005*) were used to amplify the COI region of DNA. Each PCR well contained 6.25 µl MasterMix, 4.25 µl nuclease free water, 0.5 µl forward primer, 0.5 µl reverse primer, and 1 µl DNA, for a total volume of 12.5 µl. Thermocycling occurred with an initial denaturation period of 15 min at 95 °C, followed by 35 cycles of 30 s at 94 °C, 60 s at the annealing temperature of 45 °C, and 60 s at 72 °C, followed by a final extension period of 10 min at 72 °C. PCR products were visualized using a QIAxcel system. Following PCR amplification, 2.14 µl ExoStar was added to each well to clean the PCR product. The mixture was incubated in the thermocycler for 60 min at 37 °C, then 15 min at 85 °C. Cleaned PCR products were sequenced via Sanger sequencing via ABI 3730xl sequencers in the KAUST Bioscience Core Lab.

In the event that a sample sequence did not yield a long enough strand of base pairs to be entered into a database (~500 bp), DNA was extracted again using a more precise Qiagen DNeasy Blood and Tissue kit. The PCR process was then repeated on those samples.

After sequencing was completed, sample sequences were checked against several sequence databases for potential matches. The National Center for Biotechnology Information (NCBI) GenBank (https://www.ncbi.nlm.nih.gov/genbank) and Barcode of Life Data System (BOLD) (http://www.boldsystems.org) were used as public databases. A custom (in-house) Red Sea fish sequence database was also checked (see *DiBattista et al., 2017*; *Isari et al., 2017*; *Coker et al., 2018*). A sequence was considered a good match for a species in the database if the matched sequence was 98% similar or higher. However, some species of cryptobenthic fishes can differ by fewer than 2% COI similarity (*Greenfield & Tornabene, 2014*; *Tornabene et al., 2015*). Sequence matches were then checked against visual guides and morphological keys to double-check identity. In the event of a non-matching sequence, a sample was identified to species level, or as close as possible, with the use of keys that examined morphological features. When a species was unable to be identified using the aforementioned methods, it was assigned an operational taxonomic unit (OTU) so as to still be included in the data analysis. OTUs were named for previously uploaded sequences in the databases if there was a match. In the event a sequence did not match a previously assigned OTU within one of the databases, a new OTU was assigned. COI sequences for sequenced species, including new OTUs, are deposited to GenBank under accession numbers MG583518–MG583524 and MH160733–MH160761.

## Morphological fish identification

With the use of keys, fishes that could not be identified genetically were assigned to a species, genus, or family level. The typical morphological characters that were useful in discriminating taxa included: counts of fin spines and soft rays, pelvic fin structure,

cephalic pore counts, lateral line scale counts, as well as general meristics. Morphological characters were assessed using light dissection microscopy and an online image analysis tool for measuring morphometrics (*Froese & Pauly, 2017*). For a list of keys used, see Appendix 1.

## Community analysis

Species richness was defined as the number of species or OTUs identified from each quadrat. Diversity (Shannon's diversity index, H′) was calculated using the following formula: $H' = -\Sigma \, pi \ln (pi)$ (*Shannon & Weaver, 1963*), and then subsequently averaged within each microhabitat type. Analysis of variance (ANOVA) was used to compare species richness, abundance, and diversity among habitat treatments. Significant pairings were identified using a post-hoc Tukey test. Fish communities present at each microhabitat type were plotted using non-metric multidimensional scaling (nMDS) using a square root transformation and Bray–Curtis resemblance matrix. A hierarchical cluster analysis was used to further illustrate the differences between microhabitat types. Permutational analysis of variance (PERMANOVA) was used to analyze fish community differences among microhabitats. A permutational analysis of multivariate dispersions (PERMDISP) test was performed to test for homogeneity of dispersions between microhabitat clusters. The dispersion within the three microhabitats did not differ significantly in all fish (PERMDISP, $F_{2,12} = 2.67$, $p = 0.217$), or in gobies (PERMDISP, $F_{2,10} = 3.94$, $p = 0.116$). However, pairwise comparisons between microhabitat types for all fish differed significantly between coral and rubble (PERMDISP, $p = 0.038$), and between rubble and sand in all gobies (PERMDISP, $p = 0.008$). A similarity percentage analysis was conducted to identify which fish species contributed most to the differences between microhabitats. Analyses were conducted with R version 3.4.0 (*R Core Team, 2017*), (vegan package: *Oksanen et al., 2017*), and PRIMER-e v6 (*Clarke & Gorley, 2006*).

## RESULTS

### Fish communities

A total of 326 individuals representing 73 species and 17 families were collected from three microhabitat types (Table S2). The total number of all fishes collected at each quadrat ranged from 1 to 65. Rubble had the highest average numbers of fish abundance (ANOVA $F_{2,12} = 11.59$, $p = 0.002$) and species richness ($F_{2,12} = 6.78$, $p = 0.011$), followed by coral, and then sand (Fig. 2). Diversity did not have a significant effect among microhabitats ($F_{2,12} = 2.80$, $p = 0.10$).

Quadrats that were associated with the same microhabitat type grouped more closely together than unrelated microhabitat types (Fig. 3). Fish communities in all three microhabitats differed significantly (PERMANOVA $F_{2,12} = 2.63$, $p < 0.001$) (Fig. 3A) with significant differences detected between coral and rubble ($p = 0.006$), coral and sand ($p = 0.008$), and rubble and sand ($p = 0.012$). The dendrogram of the cluster analysis supports the nMDS by showing the groupings of the three microhabitat types sampled (Fig. 3C). Coral and rubble microhabitats are more similar to each other than to the sand microhabitats (Figs. 3C and 3D).

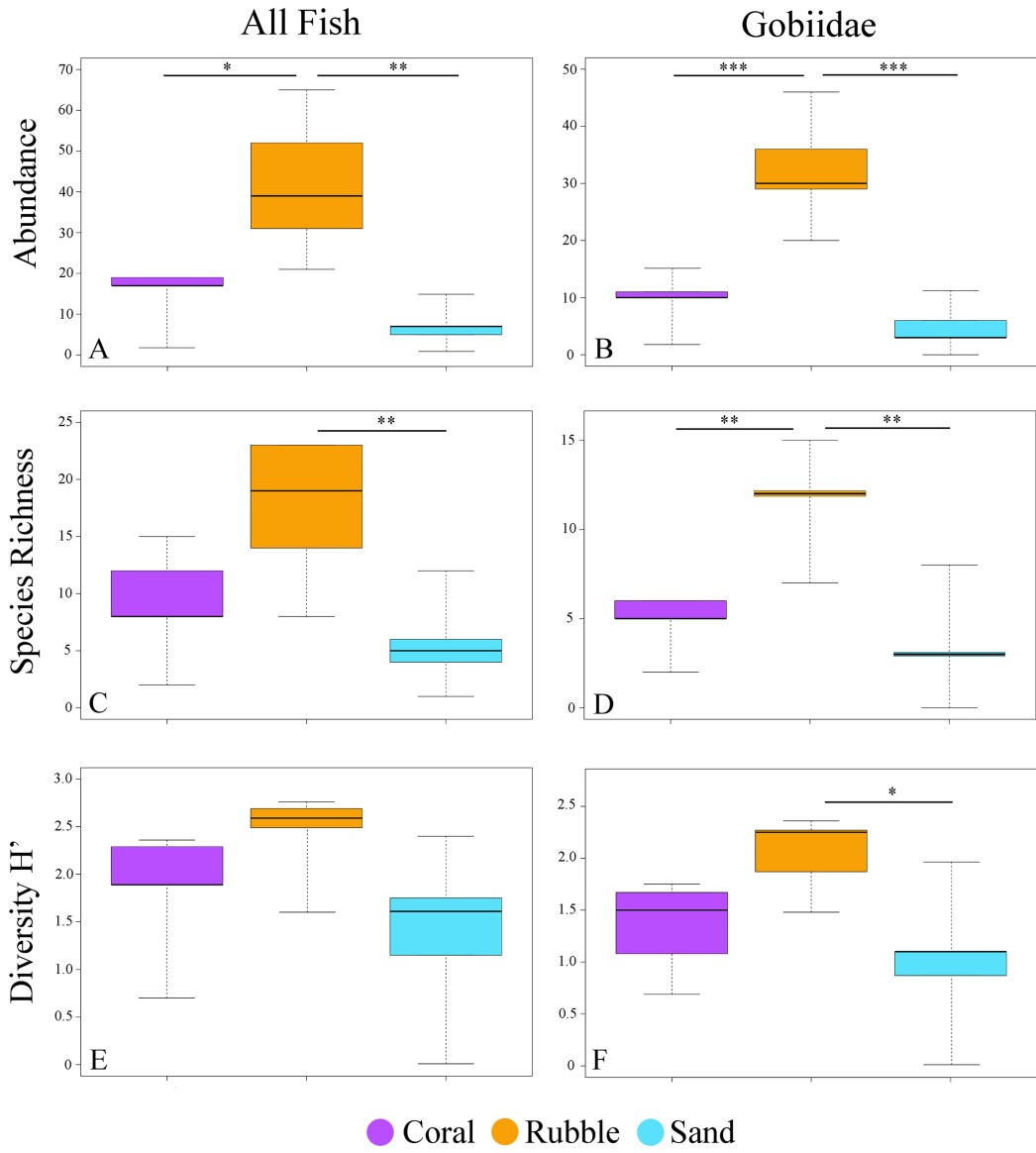

**Figure 2 Fish community indices.** Boxplots displaying abundance (number of individuals per m$^2$), species richness (number of species per m$^2$), and diversity (H′) for all collected fishes (A, C, E) and for family Gobiidae (B, D, F) for each of the three microhabitat types sampled using rotenone stations (1 m$^2$ quadrats, $n = 5$ quadrats per microhabitat type) in the central Saudi Arabian Red Sea. Asterisks denote significant pairwise groupings (*$p < 0.05$, **$p < 0.01$, ***$p < 0.001$).

## Goby communities

Out of the 326 fish collected, family Gobiidae comprised the majority, with a total of 232 individuals (71.1%) collected, representing 31 species. On average, rubble communities had the highest levels of goby abundance (ANOVA $F_{2,12} = 24.61$, $p < 0.001$), species richness ($F_{2,12} = 14.95$, $p < 0.001$), and diversity ($F_{2,12} = 5.10$, $p = 0.025$), followed by coral communities, and then sand communities (Fig. 2). Abundance ranged from 0 to 46 gobies per quadrat. The most abundant goby was *Trimma avidori*, with a total of

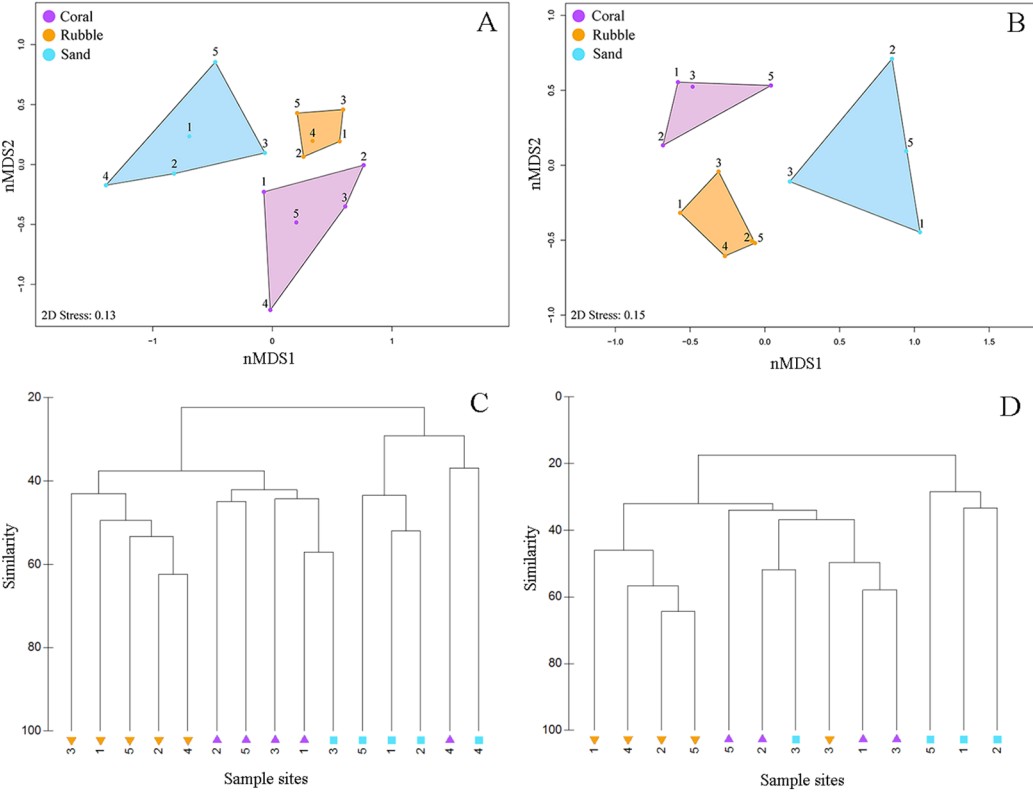

**Figure 3 Fish community similarities.** nMDS plots using Bray–Curtis similarity and square root transformations of (A) all fish communities (i.e., relative abundances of all collected species) and (B) goby communities (i.e., relative abundances of individuals from family Gobiidae) sampled using rotenone stations in each of the three microhabitat types in the central Saudi Arabian Red Sea. Solid points represent individual quadrats from each microhabitat type. Dendrograms of the hierarchical cluster analysis show habitat clustering for (C) all fish and (D) gobies. Similar microhabitat communities are more closely clustered. (Note that quadrats Coral 4 and Sand 4 were excluded from the goby nMDS and dendrogram due to an absence of gobies in those quadrats).

44 individuals collected in both coral and rubble microhabitats. The second most abundant goby was *Callogobius bifasciatus*, with 29 individuals collected in mainly rubble quadrats (Fig. 4). Of 31 goby species collected, 11 species (35.4%) were represented by only a single individual. The majority of species (64.5%) had fewer than five individuals collected.

Overall, goby community composition significantly differed among microhabitats (PERMANOVA $F_{2,12} = 3.67$, $p < 0.001$) (Fig. 3B), with more similar microhabitats grouping together (Fig. 3D). Pairwise comparisons between microhabitats show significant differences between coral and rubble ($p = 0.01$), coral and sand ($p = 0.03$), and rubble and sand ($p = 0.007$).

Gobies *C. bifasciatus*, *Asterropteryx semipunctata*, *T. avidori*, and *Eviota zebrina* contributed the most to the differences found between rubble and sand microhabitats (average contributions 12.58%, 12.53%, 8.55%, and 6.42%, respectively) (Table 1). These four species are also the highest contributors to differences found between coral and

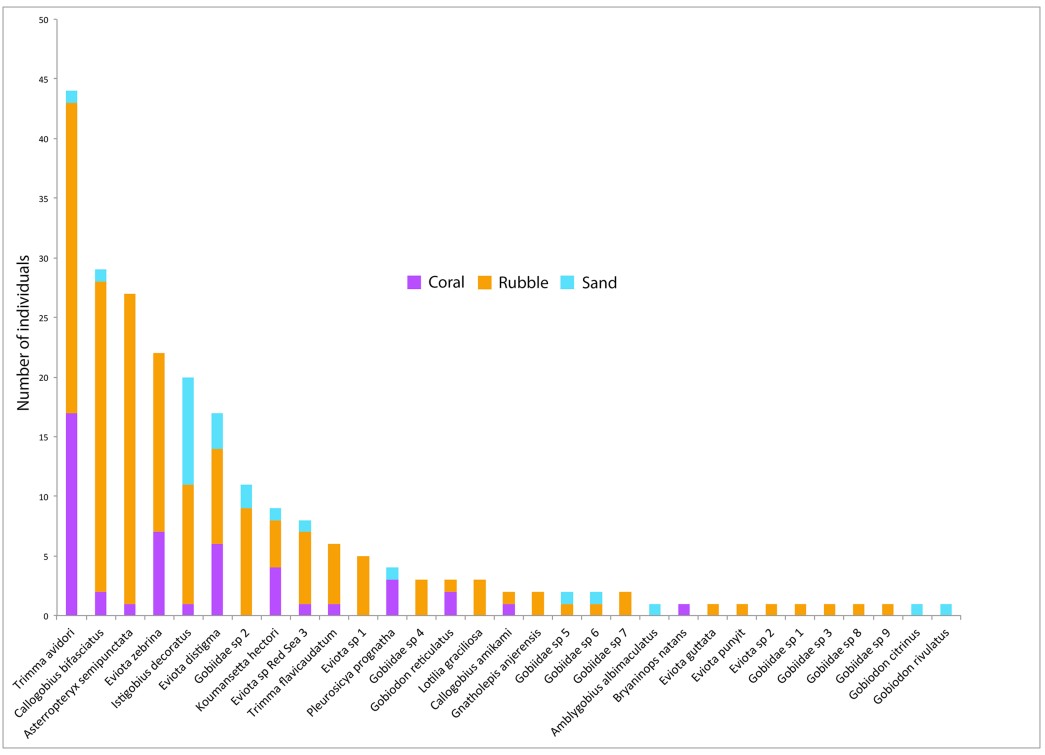

**Figure 4 Goby species abundances.** Rank-abundance plot of goby species sampled in the central Saudi Arabian Red Sea using 1 m² rotenone stations in three microhabitat types. Vertical bars represent the total combined number of individuals sampled in all quadrats ($n = 5$ quadrats per microhabitat type) for all 31 goby species found in this study. Coloration of the vertical bars indicates the portion of individuals of each species found in the three microhabitat types.

rubble microhabitats (cumulative contribution 30.54%). Between coral and sand microhabitats, *Pleurosicya prognatha* and *Istigobius decoratus* were the highest contributors (10.94% and 7.73%), with *T. avidori* following close behind (5.77%).

## DISCUSSION

This study offers insight into the scale at which cryptobenthic fishes may be influenced by habitat characteristics in the Red Sea. Within one reef, we found community differences among three microhabitats. Overall, there is evidence that rubble habitats support a greater number of individuals and species of cryptobenthic fishes. However, live coral and sand habitats may be important for more specialist species. Therefore, the majority of this group may depend on the fine scale complexity of the reef rather than benthic cover per se. The Red Sea has high rates of biodiversity and endemism (*DiBattista et al., 2016*), and this was also reflected in the relatively unexplored cryptobenthic communities with 73 species collected, of which 17 are endemic to the Red Sea (Table S2).

Cryptobenthic fishes have many habitat preferences and can be specialized or generalized to live in certain microhabitats. Food type and availability are possible drivers of specialization or generalization with this group. Certain gobies, like *Bryaninops natans* and *Pleurosicya prognatha,* were found primarily in coral dominated habitats. These fish are highly adapted coral-obligates and depend solely upon live corals and coral mucus

**Table 1 Similarity percentage (SIMPER) analysis of all collected fish species.**

**Rubble vs. Sand**

| Species | Average abundance | | Average Contribution (%) |
|---|---|---|---|
| | Rubble | Sand | |
| *Callogobius bifasciatus* | 5.2 | 0.2 | 12.58 |
| *Asterropteryx semipunctata* | 5.2 | 0 | 12.53 |
| *Trimma avidori* | 5.2 | 0.2 | 8.55 |
| *Eviota zebrina* | 3 | 0 | 6.42 |
| *Istigobius decoratus* | 2 | 1.8 | 3.37 |

**Coral vs. Sand**

| Species | Average abundance | | Average Contribution (%) |
|---|---|---|---|
| | Coral | Sand | |
| *Pleurosicya prognatha* | 2.8 | 0.2 | 10.94 |
| *Istigobius decoratus* | 0.8 | 1.8 | 7.73 |
| *Trimma avidori* | 1.4 | 0.2 | 5.77 |
| Callionymidae sp. 1 | 0.2 | 1 | 5.65 |
| Pseudochromidae sp. 2 | 1.2 | 0 | 5.17 |

**Coral vs. Rubble**

| Species | Average abundance | | Average Contribution (%) |
|---|---|---|---|
| | Coral | Rubble | |
| *Asterropteryx semipunctata* | 0.4 | 5.2 | 9.42 |
| *Callogobius bifasciatus* | 1 | 5.2 | 8.83 |
| *Trimma avidori* | 1.4 | 5.2 | 7.36 |
| *Eviota zebrina* | 0.6 | 3 | 4.93 |
| *Pleurosicya prognatha* | 2.8 | 0 | 4.66 |

**Note:**
The average contribution of the top five species contributing most to the differences between microhabitat types is listed. Average abundance (1 m² quadrats, $n = 5$ quadrats per microhabitat type) is listed for each species collected from a reef in the central Saudi Arabian Red Sea.

for food (*Herler, 2007*; *Herler, Koblmuller & Sturmbauer, 2009*). Other fish, like *Eviota distigma*, were found in almost equal abundances across all three microhabitat types. Many *Eviota* spp. are broad habitat generalists, choosing to live in a variety of habitats (*Depczynski & Bellwood, 2004*; *Herler, Munday & Hernaman, 2011*). They feed mainly upon small invertebrates, such as copepods, (*Greenfield, 2017*) that can be found across a wide variety of microhabitat types (*Preston & Doherty, 1994*; *Stella, Jones & Pratchett, 2010*).

The architecture provided by physical structures within each microhabitat provides negative spaces of different sizes that are likely to have an influence on the presence and abundance of cryptobenthic fishes. Highly complex areas offer a means to avoid and escape larger predators (*Beukers & Jones, 1997*). The importance of complex habitats is evident by positive correlations with the abundances of conspicuous fishes (*Caley & St John, 1996*; *Friedlander & Parrish, 1998*), and cryptobenthic fishes (*Depczynski & Bellwood, 2004*). High complexity, as measured by rugosity, was found in coral quadrats;

however, this course scale is likely more important for larger fishes than for cryptobenthics. Despite the coral microhabitats having very high complexity, they did not have the highest abundance levels. Rubble microhabitats had extremely high fish abundances, suggesting that measuring rugosity on a broad scale may not be relevant for small, cryptic fishes. Instead, small, intricately laid pieces of rubble, may offer enough shelter from predators for tiny fishes, and prove more important to their survival than live coral or sand habitats.

In addition to investigating habitat associations, this study helps build on the limited available information about this group through collected specimens. These provide sequence data and vouchered specimens for many unknown Red Sea species, an effort that will undoubtedly assist future studies within the region. This study provides genetic sequences of species/OTUs for the Red Sea along with photographs. However, of all the fish collected, 84 (25.7%) could not be confidently assigned to species level using either morphological or genetic techniques. One candidate for new species was identified from these OTUs, and is currently being assessed by expert taxonomists. These difficulties associated with identifying cryptobenthic fishes highlight the need for a comprehensive sequence library that includes reference images, as well as for more cryptobenthic taxonomists actively describing new species.

While this study was limited in scale, differences were observed in cryptobenthic fish communities among habitats at the scale of a reef site. Species accumulation curves (Fig. S1) suggest that increased sampling (spatially and replication) would yield more unique species. However, we expect the observed patterns among habitats to hold. Furthermore, we only investigated three common reef habitats, whereby countless more exist (e.g., soft coral, seagrass, depth, exposure), and in a gradient of combinations. Given their intimate relationship with the benthos at a small scale, we expect that communities would differ among most habitats driven by species with more specialist resource requirements. For consistency, this study targeted coral microhabitats containing *Acropora* spp. Given fishes can have species-specific coral relationships (*Munday, Jones & Caley, 1997*; *Munday, 2002*), habitats containing other coral species and morphologies would also significantly influence the presence/absence and abundance of some species. Surprisingly, three *Gobiodon* individuals and one *P. prognatha* individual were collected from a sand habitat, despite being known coral-obligate species (*Herler, 2007*). While live coral was not present in these habitats, we postulate that these originated from nearby live corals as a result of drifting rotenone in the water column and diver disturbance. Despite this, we still find clear differences among microhabitats.

This study is one of the few to examine microhabitat associations of cryptobenthic fishes in the Red Sea. These fishes are undoubtedly an important functional group, but they may hold more importance to the Red Sea ecosystem. Higher temperatures increase growth rates as well as shorten larval duration times among coral reef fishes (*Green & Fisher, 2004*). This effect would most likely be multiplied in cryptobenthic fishes due to their already fast growth rates and short lifespans (*Depczynski & Bellwood, 2006*). In the Red Sea, which is warmer than other bodies of water (*Edwards, 1987*), cryptobenthic

fishes would most likely have higher turnover rates and be able to contribute more to the food web, than their counterparts in other areas of the ocean.

## ACKNOWLEDGEMENTS

The authors would like to thank Michael Campbell for his mapmaking expertise, as well as Calder Atta, Royale Hardenstine, Alison Monroe, Tullia Terraneo, Matthew Tietbohl, and Sara Wilson for their assistance in the field. We are also grateful to Calder Atta, Simon Brandl, and Luke Tornabene for their help in identifying several fish species. Fieldwork was supported by the KAUST Coastal and Marine Resources Core Laboratory. Feedback from Christopher Goatley and Luke Tornabene greatly improved the manuscript.

### Funding

This project was funded by the King Abdullah University of Science and Technology (baseline research funds to Michael L. Berumen). The funders had no role in study design, data collection and analysis, decision to publish, or preparation of the manuscript.

### Grant Disclosures

The following grant information was disclosed by the authors:
King Abdullah University of Science and Technology.

### Competing Interests

The authors declare that they have no competing interests.

### Author Contributions

- Emily M. Troyer conceived and designed the experiments, performed the experiments, analyzed the data, prepared figures and/or tables, authored or reviewed drafts of the paper, approved the final draft.
- Darren J. Coker authored or reviewed drafts of the paper.
- Michael L. Berumen contributed reagents/materials/analysis tools, authored or reviewed drafts of the paper.

### Animal Ethics

The following information was supplied relating to ethical approvals (i.e., approving body and any reference numbers):

Fish sampling was done in accordance with the guidelines and procedures approved under the King Abdullah University of Science and Technology (KAUST) Institutional Animal Care and Use Committee (IACUC).

### Field Study Permissions

The following information was supplied relating to field study approvals (i.e., approving body and any reference numbers):

The research was undertaken in accordance with the policies and procedures of the King Abdullah University of Science and Technology (KAUST).

## DNA Deposition

The following information was supplied regarding the deposition of DNA sequences:

GenBank MG583518–MG583524 and MH160733–MH160761.

Sequence data can also be found in the Supplemental Information.

## Data Availability

The R code and raw data are provided in the Supplemental Files.

## Supplemental Information

Supplemental information for this article can be found online at http://dx.doi.org/10.7717/peerj.5014#supplemental-information.

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
