# Peer review of "Comparison of cryptobenthic reef fish communities among microhabitats in the Red Sea"

_PeerJ, doi:10.7717/peerj.5014_

## Round 0.1 · original submission · Major Revisions

This study examines the cryptobenthic fish community in five 1-meter squared samples of three microhabitat types on one section of one Red Sea reef at one time using rotenone sampling. It finds differences between the communities in those microhabitats.

Reviewer 2 indicates that the manuscript is publishable but would like additional analyses (Bray-Curtis, SIMPER, rarefaction), acknowledgement that using rotenone without restricting its dispersal likely resulted in the inclusion specimens from adjacent but different habitats, and citation of additional literature. Although Reviewer 2 indicated his concerns require only minor revision, additional analyses and interpretation as he suggested are normally considered major revision.

Reviewer 1 expresses concerns that the limited replication and spatial scale of the study makes it unable to address its goals ('examine differences in cryptobenthic fish associated with three common microhabitats . . . to better understand abundance and distribution within the region and provide baseline data about cryptobenthic fishes and their functional role'). In addition, he notes that by choosing tabulate corals, the coral microhabitat likely incorporates rubble under the coral cover. Furthermore, he has concerns about whether the data meet the assumptions of the PERMANOVA analysis and whether the stress values of the MDS are too high for reliable conclusions. Despite his doubts about the publishability of the study, Reviewer 1 indicated major revision rather than rejection.

I agree with Reviewer 2 that the scope of the study greatly limits general conclusions, I recognize that substantial effort has already gone into the data collection and analysis and that the scarcity of data on Red Sea cryptobenthic fishes makes even spatially and temporally limited data valuable. While I would recommend that the study be expanded into a more substantial publication through additional sampling, I recognize that this may not be feasible. If this is the case, it may be possible to publish a paper that more accurately recognizes the limitations of the study, avoiding claims that it provides a baseline for the study of cryptobenthic fishes in the Red Sea, for example. Therefore, I have selected a decision of 'major revisions' to give you this option, if needed.

With regard to the presentation, both reviewers have made many helpful suggestions. I agree with Reviewer 1 that the manuscript tends to be a bit verbose and redundant and with Reviewer 2 that more attention to what is known about the species found would be useful. I have provided an annotated pdf version with a number of suggestions for corrections of grammar and word choice and figure formatting, as well as a couple of other questions about the interpretation. If you choose to resubmit, please respond to the comments on pdfs by Reviewer 2 and by me. You don't have to mention grammatical and minor wording suggestions unless you disagree and are retaining your wording or style.

·

Basic reporting

The manuscript presents a study on the distribution of cryptobenthic fishes among three microhabitats on a single reef in the central eastern Red Sea. While these fishes are poorly studied, particularly in the Red Sea, and the manuscript is quite well presented, I have some serious reservations as to the rigour and technical standard of the study. These concerns are presented in the sections below.

1. The language throughout could be improved a little. It has a tendency to be too wordy and formal. Phrases like "have been found to be" and "are shown to be" should be avoided (e.g. lines 46, 54-55, 81-82 etc.). Instead, just state the finding of the paper and cite it (e.g. instead of "bananas were found to be yellow [Bloggs 1997]", write "bananas are/may be yellow [Bloggs 1997]"). A thorough proof read throughout would also help to improve a lot of little typos.

2. Paragraphs also tend to be very long (e.g. paragraph 1 of the introduction is 21 lines long and paragraph 2 is 29 lines long). Paragraphs need to present 1 idea, and as a basic rule of thumb, you should aim for most being around 10 lines long.

3. In the abstract, stating the number of fishes collected is of no use as you do not state the sampling regime. Presenting the data per unit area or similar would be more intuitive.

4. The authors mention the potential impacts of coral reef degradation in a few places in the introduction (lines 61-63, 108-109) but do not mention this idea in the discussion at all. If you are introducing this concept (particularly with regards to coral>rubble>sand being a pathway for degradation) you should consider the potential implications for reef communities in the discussion (or alternatively remove it from the introduction).

5. As a related comment the writing in lines 61-63 is very unclear. This sentence needs to be restructured. Maybe make a complete paragraph about degradation that compiles lines 61-63 and 108-109 so this concept isn't 'tacked-on' to other paragraphs with other themes.

6. In line 77-78 you make it sound like the 2 studies you cite do not account for cryptobenthic reef fishes, when they are examples of the few studies that do. Be careful with your wording and tone (you don't want to annoy other authors inadvertently).

7. Line 89-90 - Depczynski and Bellwood 2004 does not measure home ranges, you should look at Luckhurst and Luckhurst 1978a cited in D&B and draw your own conclusions on whether their home range estimates are reliable.

8. Paragraph 1 of the discussion needs to be restructured to state the findings of the study and put them in context. at the moment the 4 sentences do not link with each other at all.

9. Lines 285-onward. Be very careful with your discussion of sediment vs. detritus (these are not the same thing). Increased sediment decreases the availability of detritus. The current explanation seems to indicate a limited understanding of this concept and there are some mis-citations (Goatley et al. 2016 suggests sediments are a bad thing for cryptobenthic fishes). Perhaps think more about surface area and refuge availability in different microhabitats.

10. Line 297-298,. Thankfully the coral microhabitats (that you selected because they were dominated by corals) were dominated by corals! I don't think it's really necessary to make this point.

11. Line 316-320. What about sand dwelling gobies such as shrimp gobies and even things like Istigobius spp.? Would it be worth mentioning how far the sand was from other microhabitats/shelter?

12. Lines 327-329. If you're discussing the impacts of sediment size it would be worth including references surely? As a gratuitous cite-me request you could look at work by S. Gordon and/or S. Tebbett that consider the impacts of sediment size on feeding in fishes.

13. Line 341, do you mean a biodiversity hotspot?

Experimental design

The research is clearly in the aims and scope of PeerJ, and the aims/question are well stated in the introduction.

14. The description and rigour of the methods, however is currently very poor. The data are collected from 5 replicate samples from each of 3 microhabitats from 1 site on 1 reef. This experimental design means that the patterns observed cannot even be stated to be representative of even the leeward side of the reef (which appears to be ~10km long), let alone the entire central Red Sea. To accurately assess whether the patterns are representative of an area larger than the site studied, you really need to conduct sampling at multiple sites and include site as an extra, random factor in your analyses. If you do not find differences among sites, then you can begin to discuss the likelihood of the findings being representative of a larger area. Unfortunately, without this level of replication I cannot see how this manuscript is suitable for publication, as your data do not support your aims and conclusions.

15. I am also worried about your methods regarding sampling tabulate acroporids as your 'coral' microhabitats. The issue with tabulate corals (as I present in Goatley and Bellwood 2011) is that there is stuff underneath them (clearly visible in Fig. 1B), and your 'coral' microhabitats are therefore a composite microhabitat of the coral and whatever is beneath them. From your data I would posit that the predominant benthic cover beneath your coral canopies was rubble, which would explain why your rubble communities appear as a subset of your coral communities in Fig. 3A (i.e. why the rubble polygon is enveloped in the coral polygon). Unfortunately as you have overlap in your microhabitats this makes analyses/interpretation very difficult and if the benthos beneath the canopies differs among your samples it would be even more challenging to assess . Luckily you took photographs of each quadrat so you may be able toi account for this post-hoc. For the future I would recommend looking at 'reef' microhabitats (i.e. coral growing on matrix) so your microhabitats actually differ from each other and do not overlap.

16. Regarding the methods, you do not state what depth the samples were taken from. This is potentially very important as an environmental factor, as it relates to not only depth but also hydrodynamics, light and nutrients.

17. I really struggled to understand why you would use slow and expensive genetic based identification before morphological techniques. It seems very backwards to me. Why would you not identify your fishes to the best of your abilities, then barcode some of them as a tool to verify your findings???

18. Line 227-228. Explain which statistical tools you and why you used them in the appropriate places rather than just saying which stats you used at the end of the paragraph. As it stands, I have no idea why or what you used these techniques for.

Validity of the findings

While the study is essentially a low replication repeat of Depczynski & Bellwood 2004 conducted in the Red Sea. The authors repeatedly point out that little work has been conducted on cryptobenthic fishes in the Red Sea, and highlight the potential of finding new and important results from this study. I agree with these statements. However, the study design means that little useful information can be gleaned from the data.

19. As mentioned above, the level of replication is simply not sufficient to draw any conclusions about general trends of microhabitat utilisation. However, there are also potential statistical problems with the data, probably associated with low replication rates within the microhabitats. One of the (and pretty much the only) assumptions of PERMANOVAs is homogeneity of dispersion of the data. As PERMANOVAs consider relative positions of multivariate data clouds, not the point density in the clouds, you can have problems with your findings if the clouds are different sizes with the centroids in different positions. Using PRIMER you would do a PERMDISP to test these assumptions. Eyeballing Fig. 3A I think you may well have problems with this as the rubble polygon is much smaller than the coral one, but completely enveloped in it.

20. Also, due to the small number of data points the stress values on your MDS plots are very high. Usually 0.15 is a cut-off for whether the findings are reliable.

21. Finally in your conclusion I don;t think you can state that these findings are a "baseline" as they are really only representative of one small part of one reef.

Additional comments

I really like the subject of this manuscript, and think it has the potential to be important. Unfortunately I very strongly believe that the authors need to collect more data to make the findings useful. Hopefully they can do this as I would like to see this manuscript published in the future.

·

Basic reporting

The article cites most of the relevant literature, but I have made suggestions on the pdf where additional literature could and should be cited. I have also made many comments/questions on various parts of the manuscript where additional analyses and discussion would greatly improve the study. Overall, once the authors address the comments I made on the PDF, the paper should be suitable for publication.

Experimental design

I have suggested that the study add a dendrogram showing the bray curtis distances between all sites in the study, as well as a SIMPER analysis to highlight key species in each microhabitat. I have also commented on the use of rotenone without a plastic tarp over the quadrat, because several coral-obligates were collected on sand or rubble. Clearly they weren't there when the net when down, but instead almost certainly came off of neighboring coral heads. This should be noted and accounted for.

Validity of the findings

The findings appear to be valid, but based on very limited sample sizes. I think it needs to be shown with species accumulation curves that 5 samples per microhabitat are only scratching the surface.

Additional comments

Very happy to see more work being done on cryptobenthic fishes. The Red Sea is a big question mark in cryto knowledge (albeit there were some great works by Herler et al), and its encouraging to know folks are studying them at the detail that they deserve. I am looking forward to more great projects coming out of KAUST. Keep up the great work.

---

## Round 0.2 · accepted · Accept

Only 1 reviewer was available to look at your revision. He and I both agree that the substantial changes you made have greatly improved the manuscript. Although the reviewer would have preferred more discussion of the overlap of microhabitats between categories, I felt that the presentation of the overlap data and mention of the topic were adequate for the present paper. I read over the manuscript and found a small number of minor grammatical/spelling errors which I indicated on an attached pdf.

# ·

Basic reporting

This revised version of the manuscript is much improved over the initial submission. There are a few grammatical errors throughout (e.g. species' in the first sentence of the introduction). While the manuscript could do with a thorough proof read, I see little need for any major changes.

Experimental design

As mentioned in my previous review, i would like to see some discussion of both the relative surface areas among microhabitats (i.e. the fact that coral and rubble have large habitat areas than sand), and in particular the effects of benthic habitats beneath table corals (I appreciate the authors have made mention of this in the introduction). As an example of the importance of this, the Istigobius in live coral is just as incongruous as the Gobiodon over sand.

Validity of the findings

Apart from the experimental design issues reiterated from my previous review (mentioned above) the data seem relatively robust. The authors have made considerable efforts to rein back their interpretation and done a good job of revising the manuscript.

Additional comments

Well done with the reviews. I'd like to see a few more caveats/explanations of your findings in the discussion and a good proof read looking for grammatical issues, but apart from this your manuscript is very nearly ready to go.